# Self-Tooling Agent: Dynamically Extending Agent Capabilities through Scientific Tool Synthesis and Invocation

## Abstract

Tools are essential for defining an agent's capabilities, yet a fundamental challenge remains: general-purpose agents lack expert tools, while specialized scientific agents rely on manually-crafted toolsets that are expensive to build and do not generalize across domains. This tool creation bottleneck limits agent adaptability and performance on novel tasks. To address this challenge, we introduce the Self-Tooling Agent (STA), an agentic framework where the policy LLM learns to dynamically arbitrate between invoking existing tools and synthesizing new, specialized ones as needed. Specifically, the training dataset is generated by reverse-engineering contexts from expert tools sourced from multiple scientific agents, while a dynamic, interactive environment provides a sand-boxed space for tool execution and registration. The framework trains the policy LLM using a two-stage process: supervised fine-tuning is used for syntax learning, while reinforcement learning with a principled, multi-component reward function optimizes the LLM's strategic decision-making. Extensive evaluations on a diverse suite of benchmarks, from complex scientific QA to standard function-calling leaderboards, demonstrate that the proposed STA significantly outperforms baselines that rely on fixed toolsets, including specialized agents and powerful proprietary models. This work establishes that empowering an agent to autonomously expand its own capabilities is a critical step towards creating more adaptable and resourceful scientific agents.

## 1 Introduction

Large Language Models (LLMs) have demonstrated remarkable proficiency in complex reasoning, yet their capabilities are fundamentally constrained by their pre-trained knowledge. A powerful agentic paradigm has emerged that augments LLMs with external tools, enabling them to access real-time information and interact with external systems (Schick et al., 2023; Patil et al., 2024). While this approach works well for general-purpose tasks, it often falls short in specialized scientific fields where problems demand a high degree of domain expertise.

To bridge this gap, a new class of domain-specific agents has emerged, designed to excel in specialized scientific fields. For instance, SciToolAgent (Ding et al., 2025) is tailored for chemistry and materials science, whereas Biomni (Huang et al., 2025) is developed for biomedical research, leveraging retrieval-augmented planning and code-based execution. Notable examples include ChemAgent (Tang et al., 2025), which employs a self-improving memory system for chemistry tasks, and Virtual Lab (Swanson et al., 2025), which supports AI-human collaboration on complex research problems like protein design. The success of these systems stems from their ability to leverage curated libraries of specialized tools, enabling them to achieve expert-level performance within their respective domains.

Although these domain-specific agents are highly effective, they reveal a deeper bottleneck: the manual and laborious process of tool construction. Their sophisticated toolsets are typically built by human experts, a process that is expensive, difficult to scale, and requires deep domain knowledge that is often scarce. Furthermore, this specialization creates rigid systems where a toolset meticulously designed for bioinformatics is not transferable to another domain like chemical synthesis.

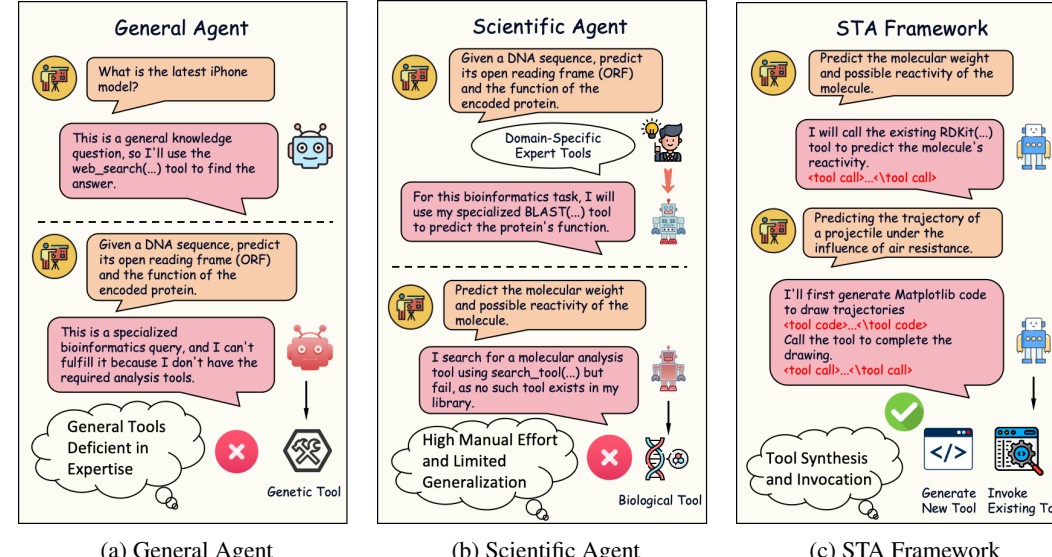

(a) General Agent      (b) Scientific Agent      (c) STA Framework

Figure 1: Comparison of agent paradigms and motivation for the proposed Self-Tooling Agent: (a) A General Agent can handle common knowledge questions using generic tools like web search but fails on specialized scientific tasks for which it lacks expert tools. (b) A Scientific Agent with a fixed, manually curated toolset can solve known domain-specific problems but cannot generalize to novel tasks that require tools outside its predefined library. (c) The Self-Tooling Agent Framework overcomes these limitations by enabling an agent to dynamically choose between invoking an existing tool (e.g., RDKit for a known chemistry task) or generating a new one on the fly (e.g., Matplotlib code for a novel plotting physical task), thus unifying tool invocation and synthesis for greater adaptability.

This lack of generalizability fundamentally limits the adaptability of scientific agents, locking expertise into isolated silos and hindering the rapid exploration of new research areas. The main challenges of existing agent frameworks are shown in Fig.1.

In response to these challenges, our **core idea** is to solve this tool creation bottleneck by empowering the agent to autonomously generate the specialized tools it needs on the fly, unifying this capability with the ability to invoke existing tools. This concept builds upon recent efforts that have explored the automated generation of tools to alleviate the human burden (Cai et al., 2023). However, these approaches often treat tool creation and tool usage as distinct, decoupled problems. They may focus on synthesizing a tool from a natural language description but fail to integrate this process into the agent's core decision-making loop. This separation leaves a critical gap, where an agent that cannot dynamically reason about when to create a new tool versus when to use an existing one remains handicapped and unable to truly adapt to novel problems.

To address this critical gap, we introduce Self-Tooling Agent, a unified agentic training framework that endows an LLM with the ability to dynamically arbitrate between generating a new, specialized tool and invoking an existing one. We make the following key contributions:

- **Unified Generation-Invocation Framework:** We formulate the agent tooling problem as a unified generation-or-invocation decision and propose Self-Tooling Agent, a novel agentic framework that empowers a model to learn this choice.

- **Dual-Task Tooling Dataset:** We construct a comprehensive dataset tailored for this dual task, providing a rich signal for learning both when and how to generate new tools and when to rely on existing ones.

- **Multi-Component Reward:** We develop a principled reward design that considers the necessity of tool generation, execution success, and final task completion, and we demonstrate the successful application of GRPO to achieve stable and effective training.

We evaluate the proposed method on a suite of benchmarks designed to test both on-the-fly tool generation, using scientific QA datasets like HLE (Phan et al., 2025) and LabBench (Laurent et al., 2024), and tool invocation, using standard benchmarks such as BFCL(Patil et al.). Experimental results consistently show that the proposed method significantly outperforms strong baseline agents. These findings validate our unified training framework, demonstrating that an agent can learn to strategically arbitrate between creating new tools and invoking existing ones to solve complex problems.

## 2 RELATED WORK

### 2.1 GENERAL AGENTS

Equipping Large Language Models (LLMs) with external tools represents a promising research frontier (Shen et al., 2024; Lu et al., 2024; Yuan et al., 2024b; Shen, 2024). This paradigm empowers models to transcend their inherent limitations by accessing external knowledge bases and executing complex actions. Despite this potential, the prevailing focus of current research remains on general-purpose tools, such as web search engines (Koh et al., 2024) and calculators (Xiao et al., 2024). Such tools, while useful, are insufficient for addressing complex scientific inquiries. For instance, they cannot devise a synthesis plan for a chemical compound with desired properties (Bran et al., 2023; Darvish et al., 2025), nor can they analyze vast genomic sequences using specialized bioinformatics software (Gridach et al., 2025; Ren et al., 2025). These tasks are essential for scientific research and require agents skilled in working with specialized computational tools.

Concurrently, a significant paradigm shift is underway, transitioning from scaling static models to developing autonomous agents capable of self-evolution (Hu et al., 2024; Tao et al., 2024). Prevailing research has primarily focused on enhancing the agent's intrinsic, model-based capabilities. For instance, InternAgent (Team et al., 2025) constructs multi-agent systems to automate and accelerate scientific research. Similarly, EvoAgent (Yuan et al., 2024a) leverages evolutionary algorithms to achieve self-improvement. While this line of inquiry has unlocked significant potential by targeting the agent's core model, it has largely overlooked another critical dimension of agent capability. Specifically, the evolution of an agent's tools has remained a largely underexplored area of research.

### 2.2 SCIENTIFIC AGENTS

The advancement of scientific agents critically depends on curated libraries of specialized tools to execute domain-specific tasks. For example, Biomni (Huang et al., 2025) leverages a unified action space of biomedical tools to dynamically compose workflows for diverse research problems. Similarly, (Zhang et al., 2025) acts as a virtual disease biologist by aggregating over 600 distinct bioinformatics tools. However, manual curation and development of these tools create a bottleneck, being resource-intensive and requiring substantial expertise to validate. Furthermore, tool libraries often lack generalizability, as they are custom-built for specific domains, limiting adaptability and hindering the scalability and impact of agent-based systems in scientific discovery (Team et al., 2025; Wang et al., 2024a; Weil et al., 2024).

### 2.3 AGENT TOOL SYNTHESIS

Automated tool generation has recently emerged as a compelling alternative to manual development, leveraging the generation capabilities of LLMs to synthesize new tools. For instance, CodeAgent (Zhang et al., 2024) utilizes prompts to guide LLMs in searching online resources and generating corresponding tool functions, while the framework by (Wang et al., 2024b) reframes tool interaction itself as a generative task. Still, applying these approaches to scientific problems exposes a critical limitation: general-purpose LLMs lack the requisite domain-specific knowledge. Consequently, the tools they generate are often functionally incorrect to the nuances of scientific inquiry (Chen et al., 2025; Nejjar et al., 2025). Moreover, existing generation pipelines are frequently rigid, as they are designed exclusively for tool creation, which precludes their direct application to downstream tasks such as complex question answering (Wang et al., 2024c). To address these deficiencies, we introduce Self-Tooling Agent, a unified agentic framework that integrates tool generation and invocation in an end-to-end reasoning process, allowing the agent to dynamically create and apply tools for the given scientific task.

## 3 PRELIMINARY

In this section, we first formulate the unified agentic tool generation and invocation problem in Section 3.1. We then discuss the high-level design considerations of our Self-Tooling Agent framework in Section 3.2. For the core concept of this section, we provide an example explanation in Fig2.

### 3.1 PROBLEM FORMULATION

**Agentic Tool Interaction as a Decision Process.** We conceptualize the challenge of dynamic tool augmentation as a sequential decision-making process. An agent, given a complex task, must iteratively decide on the optimal action to progress towards a solution. This process is defined by the agent's current state, the actions available to it, and the overarching goal of maximizing task success. Specifically, we define the state at any step $t$ as:

$$s_t = (T, H_t, \mathcal{F}_t),\qquad(1)$$

where $T$ is the initial task description, $H_t$ is the history of previous actions and observations, and $\mathcal{F}_t$ is the set of available tools at that step. The agent's goal is to learn an optimal policy $\pi^*\left(a_t \mid s_t\right)$ that selects an action $a_t$ to maximize the expected cumulative reward over a trajectory, ultimately leading to the successful completion of task $T$. The key distinction from prior work is that the toolset $\mathcal{F}_t$ is not static; it can be augmented during the process, such that $\mathcal{F}_{t+1} = \mathcal{F}_t \cup \{f_{\text{new}}\}$ if a new tool $f_{\text{new}}$ is generated and executed successfully.

**The Generation-Invocation Action Space.** Central to our formulation is a constrained and meaningful action space, $\mathcal{A}$, where the agent's decision is manifested through its generated output. The LLM is trained to produce one of two distinct, structured responses enclosed in special tokens:

- **<tool_code>...<\tool_code>:** Generating content within these tags signifies the decision to synthesize a new tool. The enclosed content is the tool's source code, typically a Python function, which is then added to the agent's available toolset $\mathcal{F}_t$ for subsequent steps.

- **<tool_call>...<\tool_call>:** Generating content within these tags signifies the decision to invoke an existing tool. The enclosed content specifies the tool's name and the parameters for its execution, selected from the set $\mathcal{F}_t$.

This binary space of action over tools forces agents to reason explicitly about the trade-offs between leveraging existing capabilities and investing in new ones.

### 3.2 METHOD OVERVIEW

**STA Framework and Execution Environment.** The proposed STA is a holistic framework designed to train an LLM to master the unified generation-or-invocation task. The core of our approach is a policy LLM that is trained within a dynamic interactive environment, and the overview of the proposed method is shown in Fig2. The training proceeds in two stages: Supervised Fine-Tuning (SFT) and Grouped Relative Policy Optimization (GRPO). First, we fine-tune the LLM with parameters $\theta$ to learn the syntax of tool generation and invocation. The SFT objective is to minimize the negative log-likelihood of expert actions from our dataset $D$:

$$\mathcal{L}_{\text{SFT}}(\theta) = -\sum_{(s,a)\in\mathcal{D}} \log \pi_\theta(a|s).\qquad(2)$$

To move beyond simple imitation and learn a robust decision-making policy, the LLM must learn from the consequences of its actions. This is facilitated by our interactive environment, which is based on mini-swe-agent (Yang et al., 2024). The environment serves as a sand-boxed execution and storage space where the agent can use 'bash' commands to interact with specialized scientific resources and where newly generated tools are dynamically registered. This setup enables the GRPO training phase, where the policy is refined to optimize for task success. The objective of this stage is to find the policy that maximizes the expected reward $R$ over a full task trajectory $\tau$:

$$\max_\theta \mathbb{E}_{\tau\sim\pi_\theta}[R(\tau)].\qquad(3)$$

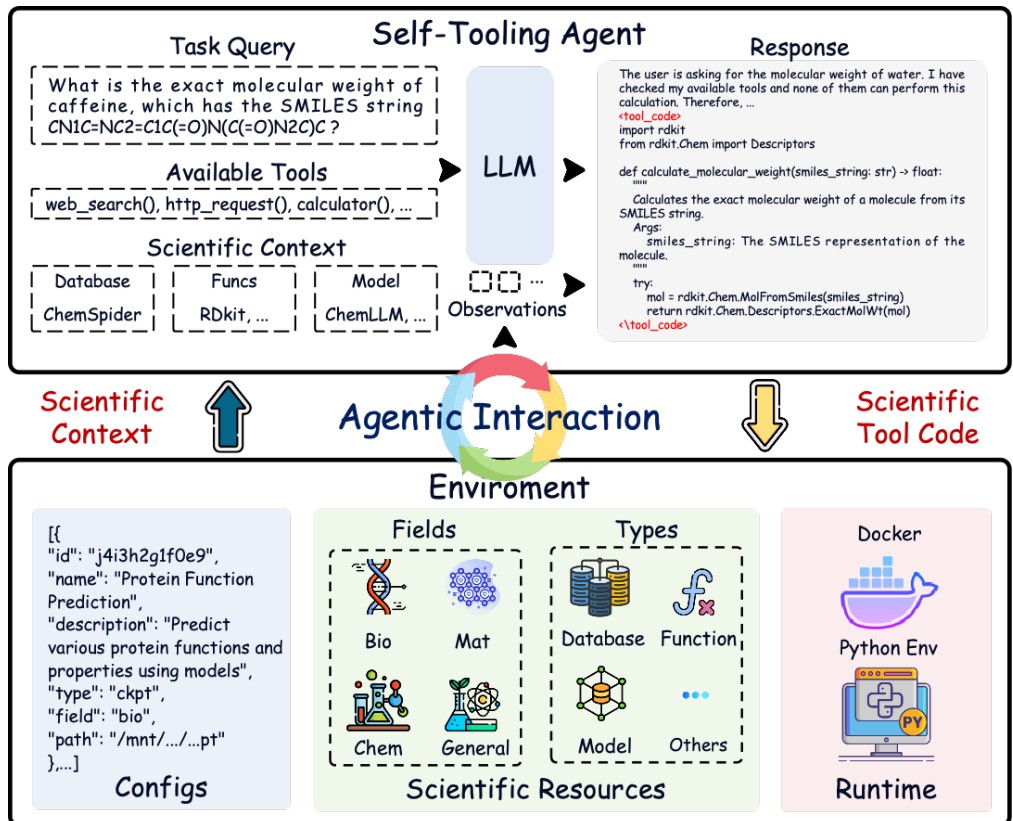

Figure 2: Overview of the Self-Tooling Agent framework. The figure illustrates the agent's workflow for a specialized scientific task. When a Task Query (e.g., calculating molecular weight) cannot be addressed by the initial Available Tools, the core LLM plans to create a new tool. This process occurs within the central Agentic Interaction loop, which is grounded in an Environment that provides rich Scientific Context, including access to libraries (like RDKit) and other scientific resources. The LLM leverages this context to generate new Scientific Tool Code, demonstrating the framework's core capability of dynamically synthesizing tools to solve problems beyond its initial toolkit.

**Task Scope.** To comprehensively evaluate the proposed method, we ground our experiments in two distinct types of benchmarks. First, to evaluate performance on complex, domain-specific problems, we utilize a suite of challenging science QA benchmarks, including HLE (Phan et al., 2025), LabBench (Laurent et al., 2024). These benchmarks are designed such that many questions are intractable without access to specialized tools for calculation or data retrieval, making them an ideal testbed for assessing the agent's dynamic tool generation capabilities. Second, to evaluate the agent's proficiency in using existing tools, we employ a standard tool call evaluation benchmark. This benchmark tests the model's ability to select the correct tool from a given library and format the arguments accurately, thereby measuring the effectiveness of its invocation policy.

## 4 METHODOLOGY

This section details the core components of our Self-Tooling Agent framework. We begin by describing the novel data curation process designed to teach both tool generation and invocation in Section 4.1. We then introduce the interactive environment that facilitates dynamic tool execution and state management in Section 4.2. Finally, we detail the two-stage model training paradigm in Section 4.3.

---

**Algorithm 1** Agentic Interaction

---

**Require:** Query $Q$, Policy LLM $\pi_{\theta*}$, Environment $E$, Initial Tools $\mathcal{F}_0$, Max Iterations $K$
**Ensure:** Final Response $A_{\text{final}}$
 1: Initialize state $s_0$ with $Q$ and $\mathcal{F}_0$; $k \leftarrow 0$; Terminated $\leftarrow$ **false**
 2: **while** $k < K$ **and not** Terminated **do**
 3:     $k \leftarrow k + 1$
 4:     Policy LLM $\pi_{\theta*}$ generates action $a_k$ based on state $s_{k-1}$
 5:     **if** $a_k$ is a `<tool_call>` **then**
 6:         $o_k \leftarrow E.\text{execute\_tool}(a_k)$                                    ▷▷ Invoke an existing tool
 7:         $s_k \leftarrow \text{UpdateState}(s_{k-1}, a_k, o_k)$
 8:     **else if** $a_k$ is a `<tool_code>` **then**
 9:         Let $C$ be the code within $a_k$
10:         $o_k \leftarrow E.\text{execute\_code}(C)$                          ▷▷ Execute the newly generated tool
11:         **if** $o_k$ indicates success **then**
12:             $E.\text{register\_tool}(C)$                                    ▷▷ Save the successful tool
13:             $A_{\text{final}} \leftarrow M_{\theta}^{*}.\text{generate\_response}(o_k)$
14:             Terminated $\leftarrow$ **true**
15:         **else**
16:             $s_k \leftarrow \text{UpdateState}(s_{k-1}, a_k, o_k)$          ▷▷ Pass error to the agent for revision
17:         **end if**
18:     **else if** $a_k$ is a `<response>` **then**
19:         $A_{\text{final}} \leftarrow$ content of $a_k$
20:         Terminated $\leftarrow$ **true**
21:     **end if**
22: **end while**
23: **return** $A_{\text{final}}$

---

## 4.1 DATASET CONSTRUCTION

**Scientific Tool Generation.** To teach the model how to create new tools, we curated a dataset where the input is a rich contextual prompt and the target output is a complete, functional tool enclosed in `<tool_code>...</tool_code>` tags. Our process begins by sourcing high-quality, human-validated Python functions from the SciToolAgent and Biomni. For each function, we then reverse-engineer a realistic problem scenario that would necessitate its creation. This involves synthesizing a detailed prompt containing a clear problem statement along with the necessary background context, such as database schema or the function signatures of other relevant scientific models. This methodology produces high-fidelity (context, tool_code) pairs that teach the model to generate accurate tool code from complex, real-world requirements.

**Tool Use Trajectory.** To teach the model how to proficiently use tools, we constructed a composite dataset of invocation trajectories. The majority of these trajectories are adapted from existing, publicly available reinforcement learning (RL) datasets for tool use, providing a broad foundation of invocation patterns. To supplement this, we generated a smaller, targeted set of trajectories through an interactive, model-in-the-loop procedure(Wang et al., 2024c; Qin et al., 2023; Schick et al., 2023). In this setup, an agent solves tasks specifically designed to require interaction with our scientific execution environment. These newly generated trajectories ensure the model learns to handle the unique tools and constraints of our framework. More details on the data curation process and statistics can be found in Appendix A.2.

## 4.2 INTERACTIVE ENVIRONMENT

The interactive environment is a critical component of our framework, providing the necessary infrastructure for both the RL stage of training and the final inference. It is designed to execute scientific tools, dynamically manage the set of available tools, and provide a grounded interface to specialized scientific resources.

**Core Components.** Our interactive environment is a workspace designed to support the Self-Tooling Agent, defined by three core components. First, it is provisioned with a rich set of Scientific

Resources, including databases, models, and functions spanning fields like biology, chemistry, and materials, which are described by structured Config files. Second, a secure Runtime, consisting of a containerized Docker and Python environment, is used to execute all agent-generated code and bash commands. Finally, any new tools created by the agent are dynamically registered within this runtime, making them immediately available for interacting with the scientific resources.

**Agentic Interaction.** The agentic interaction of policy LLM with the environment follows the formal, iterative process detailed in Algorithm 1. At each turn k, the agent receives the current state $s_{k-1}$ and generates an action $a_k$ by sampling from its learned policy $\pi_{\theta^*}$. The environment $E$ processes this action and returns an observation $o_k$. The state is then updated for the next turn:

$$s_k = \text{UpdateState}\left(s_{k-1}, a_k, o_k\right) \text{ where } o_k = E\left(a_k\right). \tag{4}$$

A key feature of this interaction is the dynamic expansion of the agent's capabilities. If the policy LLM generates new tool via a action `<tool_code>` and its execution is successful, the environment registers the tool, updating the set of available tools $F$ for all subsequent steps in the episode. This loop of generating an action, receiving environmental feedback, and updating the state continues until the task is completed.

### 4.3 MODEL TRAINING

The training process refines the agent's core LLM using a two-stage approach. First, SFT establishes a baseline policy. This policy is then optimized for task success using GRPO, which enhances the agent's overall ability to create and invoke tools effectively.

**SFT for Syntax Understanding.** The primary objective is to familiarize the LLM with the specific syntax of our framework, teaching it to generate well-formed `<tool_code>` blocks and `<tool_call>` commands from the expert trajectories in our datasets. While effective for learning structure, relying on SFT alone can result in imitative reasoning that limits the model's ability to generalize to new scenarios. The SFT objective is to train the LLM. This provides the LLM with a strong initial policy before the RL phase.

**RL for Decision-Making.** While SFT teaches the LLM the syntax of tool use, it often results in imitative reasoning that fails to generalize to novel scenarios. To develop a truly agentic capability, we refine the policy LLM using reinforcement learning to optimize for strategic decision-making and task success. The RL process is guided by a multi-component reward function, $R_{\text{total}}$, which evaluates the outcomes of the agent's actions:

$$R_{\text{total}} = R_{\text{necessity}} + R_{\text{execution}} + R_{\text{completion}}, \tag{5}$$

where $R_{\text{necessity}}$ penalizes the creation of redundant tools, $R_{\text{execution}}$ execution rewards syntactically and functionally correct actions, and $R_{\text{completion}}$ provides a large terminal reward for solving the overall task. Besides, regular $R_{format}$ and $R_{correct}$ for tool use competency training are also included(Gao et al., 2025; Schick et al., 2023). More details of the reward design are in Appendix A.3.

## 5 EXPERIMENTS

### 5.1 EXPERIMENTAL SETUP

**Dataset.** The evaluation is conducted on three diverse sets of benchmarks to test both scientific reasoning and general tool-use capabilities. First, the Humanity's Last Exam (HLE) benchmark Phan et al. (2025) is used to assess performance on complex scientific questions, with text-only subsets for Chemistry (101 questions), Biology (222 questions), and Physics (202 questions). Second, we use Lab-Bench(Laurent et al., 2024), which is divided into tasks for Database Question Answering (DbQA) and Sequence Question Answering (SeqQA), to test the agent's ability on specialized bioinformatics tasks. Third, to measure general-purpose function calling, we use the Berkeley Function Calling Leaderboard (BFCL) (Patil et al., 2024), evaluating performance on both Single-Turn and Multi-Turn scenarios.

**Baselines.** The proposed method is compared against a comprehensive set of baselines tailored to each benchmark. On HLE(Phan et al., 2025), baselines include web-search agents such as Miromind

Table 1: Performance comparison across different methods and tool types on Chemistry (101), Biology (222), and Physics (202) subset of HLE benchmarks.

| Method | Tool Type | Model | Chem (101) | Bio (222) | Phy (202) |
|---|---|---|---|---|---|
| Miromind(Li et al., 2025) | Web | Qwen3-8B | 8.9% | 9.5% | 2.0% |
| Xmaster(Chai et al., 2025) | Web | Qwen3-8B | 2.0% | 6.7% | 4.0% |
| *w/o workflow* | | | | | |
| LLM | Web | Qwen3-8B | 2.0% | 9.0% | 3.5% |
| LLM+ReAct(Yao et al., 2023) | Web | Qwen3-8B | 2.0% | 7.2% | 4.0% |
| STA(Ours) | Sci | STA-8B | 3.9% | 8.3% | 5.5% |
| STA(Ours) | Sci+Web | STA-8B | **7.9%** | **12.6%** | **9.4%** |

Table 2: Performance comparison of different methods on DbQA and SeqQA subset of Lab-bench

| Method | Tool Type | Model | DbQA | SeqQA |
|---|---|---|---|---|
| LLM | None | GPT-4o-2024-05-13 | 52.3% | 52.3% |
| ReAct(Yao et al., 2023) | Code | GPT-4o-2024-05-13 | 40.8% | 81.8% |
| Biomni(Huang et al., 2025) | Sci | GPT-4o-2024-05-13 | 74.4% | 81.9% |
| STA (Ours) | Sci | STA-8B | 71.2% | 78.3% |
| STA (Ours) | Sci+Web | STA-8B | **76.9%** | **82.7%** |

(Li et al., 2025) and Xmaster (Chai et al., 2025) and standard prompting methods like a Base LLM and LLM+ReAct (Yao et al., 2023). On Lab-Bench(Laurent et al., 2024), the comparison is against a powerful proprietary LLM, GPT-4o, a ReAct (Yao et al., 2023) agent, and the state-of-the-art specialized biomedical agent, Biomni(Huang et al., 2025). On the BFCL, the STA is benchmarked against top-performing proprietary models, including Gemini-2.5-Pro and GPT-5, and strong open-source models like Qwen3-14B and Qwen3-8B. Two versions of the agent are evaluated: one with only scientific tools, denoted STA-Sci, and one with both scientific and web-search tools, denoted STA-Sci+Web.

**Implementation Details.** All of our models are built upon the Qwen3-8B foundation model. The SFT stage is implemented using the LLaMA-Factory framework, while the GRPO stage is conducted using the Verl reinforcement learning library. For the SFT phase, we fine-tune the LLM for 3 epochs with a learning rate of $2 \times 10^{-5}$. For the GRPO phase, we use a learning rate of $1 \times 10^{-6}$ and a batch size of 128.

**Metrics.** We evaluate performance using the standard metric for each benchmark. For the scientific QA benchmarks, including HLE and LabBench, we report the final Answer Accuracy. For the tool-use benchmarks, BFCL-v3, we report the Tool-Call Accuracy, which measures the correctness of the generated tool name and arguments, following the official evaluation protocols of the benchmarks.

## 5.2 EXPERIMENTAL RESULTS AND ANALYSIS

**Main Results.** The Self-Tooling Agent (STA) demonstrates superior performance on scientific QA benchmarks, validating its dynamic tool generation strategy. As shown in Table 1, the STA is the top performer across all subsets of the HLE benchmark. Furthermore, on the Lab-Bench benchmark (Table 2), the STA surpasses all baselines, including the highly specialized Biomni agent, on both the DbQA and SeqQA tasks. These results highlight the effectiveness of combining dynamic tool generation with general-purpose web search to solve complex scientific problems.

On the tool-use BFCL benchmark (Table 3), STA-8B establishes itself as a top-performing function-calling model. It achieves the highest overall accuracy in both Single-Turn (89.76%) and Multi-Turn (31.12%) scenarios, outperforming powerful proprietary models and larger open-source models. The strong performance, particularly in complex multi-turn tasks, indicates that our two-stage training process not only teaches the LLM to generate tools but also refines its ability to strategically invoke them. These combined results validate our central hypothesis that a unified framework for tool generation and invocation leads to a more capable and adaptable agent.

**Ablation Study.** To understand the impact of our reward design, we conducted an ablation study on the key components of the tool generation reward (Table 4). The results reveal that each component

Table 3: Performance comparison of different methods on Berkeley Function Calling Leaderboard (BFCL). Results show accuracy scores across single-turn and multi-turn function calling tasks. Best results in each column are highlighted in bold.

| Method | Single Turn | | | | | Multi Turn | | | | |
|---|---|---|---|---|---|---|---|---|---|---|
| | Overall | Simple | Multiple | Parallel | Multiple Parallel | Overall | Base | Miss Func | Miss Param | Long Context |
| Gemini-2.5-Pro (FC) | 85.04 | 68.67 | 91.00 | 91.50 | 89.00 | 25.00 | 25.50 | 26.00 | 24.50 | 24.00 |
| GPT-5-2025-08-07 (FC) | 72.92 | 58.67 | 76.00 | 84.00 | 73.00 | 28.50 | 33.50 | 29.50 | 23.00 | 28.00 |
| GPT-4o-mini-2024-07-18 (FC) | 80.40 | 70.58 | 87.50 | 86.00 | 77.50 | 30.12 | **43.50** | 13.50 | 26.00 | **37.50** |
| Qwen3-14B(Prompt) | 88.67 | 76.17 | 94.50 | **93.00** | 91.00 | 31.12 | 37.00 | **36.00** | 23.00 | 28.50 |
| Qwen3-8B(Prompt) | 88.60 | 78.92 | **95.00** | 91.50 | 89.00 | 24.12 | 26.00 | 29.00 | 20.50 | 21.00 |
| STA-8B | **89.76** | **80.04** | 94.50 | 93.50 | **91.00** | **31.12** | 29.50 | 32.50 | **25.00** | 35.50 |

Table 4: Ablation study results showing the impact of different reward components.

| Method | $R_{\text{necessity}}$ | $R_{\text{execution}}$ | $R_{\text{completion}}$ | Tool Use | | Scientific QA | |
|---|---|---|---|---|---|---|---|
| | | | | Single Turn | Multi-Turn | DbQA | SeqQA |
| Tool-use Reward | ✗ | ✗ | ✗ | 80.1 | 26.5 | 10.2 | 13.4 |
| w/o Completion Reward | ✓ | ✓ | ✗ | 76.3 | 76.3 | 22.9 | 30.2 |
| w/o Execution Reward | ✓ | ✗ | ✓ | 81.2 | 28.9 | 70.4 | 78.5 |
| w/o Necessity Reward | ✗ | ✓ | ✓ | 24.5 | 8.2 | 50.2 | 73.6 |
| STA | ✓ | ✓ | ✓ | **89.8** | **30.6** | **76.9** | **82.7** |

is critical for robust performance. Removing the terminal $R_{\text{completion}}$ reward impairs the agent's ability to solve complex Scientific QA problems, while the absence of the $R_{\text{necessity}}$ penalty causes a catastrophic failure on tool-use benchmarks, as the agent incorrectly attempts to generate tools instead of invoking them. Finally, removing the dense $R_{\text{execution}}$ reward weakens the agent's capacity for complex, multi-turn interactions. These findings confirm that the combination of rewards for goal-direction $R_{\text{completion}}$, efficiency $R_{\text{necessity}}$, and correctness $R_{\text{execution}}$ is essential for creating a versatile and high-performing Self-Tooling Agent.

# 6 CONCLUSION

In this work, we addressed a critical bottleneck in the development of specialized agents: their reliance on static, manually-created toolsets that are expensive to produce and lack generalizability across domains. We introduced Self-Tooling Agent, a unified agentic framework where the core LLM is trained to dynamically arbitrate between generating new, specialized tools and invoking existing ones. Our methodology is centered on three key contributions: a novel dataset construction process that reverse-engineers realistic contexts for tool generation; a dynamic, interactive environment that facilitates tool execution and state management; and a two-stage training paradigm using SFT and GRPO with a principled, multi-component reward function.

Our experiments demonstrate that the Self-Tooling Agent significantly outperforms strong baselines on both scientific QA and standard tool-use benchmarks. These results validate our central hypothesis that an agent's effectiveness is greatly enhanced when its policy LLM learns to strategically decide between creating new capabilities and leveraging existing ones. This work represents a significant step toward creating more autonomous, adaptable, and resourceful agents that can expand their own functional repertoires to meet the demands of novel and complex tasks. Future work may explore extending this framework to allow the agent to not only generate but also compose and modify existing tools, further enhancing its flexibility and problem-solving ability.

# 7 REPRODUCIBILITY STATEMENT

All authors have reviewed the final manuscript and take full responsibility for its content and claims. We are committed to ensuring the full reproducibility of our research. Upon acceptance, we will release the complete codebase and all associated resources through a public GitHub repository to facilitate further research and verification by the community.

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

# A APPENDIX

## A.1 USE OF LLMS

In the preparation of this manuscript, the LLM was used as an assistive tool to enhance the quality and clarity of the paper. Specifically, the LLM was employed for two main tasks: writing polishing, which involved refining grammar, improving sentence structure, and ensuring a consistent academic tone; and LaTeX code modifications, which included formatting, and debugging LaTeX for the algorithms and Tables presented. All content and LaTeX code generated or modified by the LLM were thoroughly reviewed and verified by the authors for scientific accuracy and correctness.

## A.2 DETAILS OF DATASET

The complete dataset used for training the proposed STA consists of 7,000 instances, carefully curated and partitioned for SFT and RL stages. The foundation of our dataset is a rich collection of expert-written scientific tools and resources sourced from established, open-source scientific agents, including Biomni(Huang et al., 2025), SciToolAgent(Chen et al., 2025), and ScienceStar(Wang et al., 2025). These sources provide a strong and diverse foundation, covering specialized domains such as biology, chemistry, and materials science.

**SFT Dataset (1,000 instances).** The SFT dataset is designed to provide the policy LLM with a foundational understanding of the required syntax and action formats before the more complex RL stage. It contains 1,000 high-quality examples, composed of a balanced mix of both tool generation and tool invocation instances, ensuring the model is exposed to both core capabilities from the outset.

**RL Dataset (6,000 instances).** The dataset for the GRPO stage is significantly larger and is divided into two specialized subsets:

- **RL for Tool Generation (3,000 instances):** This dataset is used to train the LLM's ability to synthesize new tools. The data was constructed using our reverse-engineering methodology. For each expert tool curated from the source agents, we synthesized a realistic problem context, including necessary background information and a specific problem statement, creating high-fidelity (context, tool_code) pairs.
- **RL for Tool Use (3,000 instances)**: This dataset trains the LLM's invocation policy. It consists of complex, multi-step trajectories generated through an interactive, model-in-the-loop process. This rollout method produces rich examples of sequential reasoning and tool execution, teaching the LLM how to effectively chain tool calls to solve problems.

### A.3    DETAILS OF REWARD FUNCTION

The reinforcement learning stage is guided by a carefully designed reward function that provides a learning signal to the policy LLM. The total reward for a trajectory, $R(\tau)$, is a sum of dense, per-step rewards and a sparse, terminal reward. For each action $a_t$ taken by the agent at step $t$, a step-wise reward is calculated as:

$$R_t = R_{\text{execution}}(a_t) + R_{\text{necessity}}(a_t) \tag{6}$$

The final reward for the complete trajectory is the sum of these step-wise rewards plus a final task completion bonus:

$$R(\tau) = \left(\sum R_t\right) + R_{\text{completion}}(\tau). \tag{7}$$

**Execution Reward $R_{\text{execution}}$** This component provides a dense, binary signal for the functional correctness of each action taken by the agent. It draws inspiration from rule-based reward schemes that have been shown to be effective for tool-use tasks. The reward is defined as:

$$R_{\text{execution}}(a_t) = \begin{cases} 1 & \text{if } a_t \text{ executes successfully in the environment} \\ 0 & \text{otherwise} \end{cases} \tag{8}$$

An action is considered successful if a `<tool_code>` block compiles and runs without error, or if a `<tool_call>` is made with a valid tool name and correctly formatted arguments.

**Necessity Reward ($R_{\text{necessity}}$).** Unique to our framework, this signal penalizes the policy LLM for inefficient decisions. Specifically, it discourages the generation of a new tool when a suitable one already exists in the agent's tool registry, $\mathcal{F}_{t-1}$. The reward is defined as:

$$R_{\text{necessity}}(a_t) = \begin{cases} -1 & \text{if } a_t \text{ is } \langle\text{tool\_code}\rangle \text{ and IsRedundant}(C, \mathcal{F}_{t-1}) \\ 0 & \text{otherwise} \end{cases} \tag{9}$$

**Task Completion Reward ($R_{\text{completion}}$).** This is a sparse, high-magnitude terminal reward that incentivizes the agent to achieve the overall goal of the task. It is awarded only at the end of a trajectory, $\tau$:

$$R_{\text{completion}}(\tau) = \begin{cases} 3 & \text{if the agent's final answer is correct} \\ 0 & \text{otherwise} \end{cases} \tag{10}$$

Our reward components are integrated into the GRPO training loop to update the policy LLM. For each prompt, the policy LLM generates a group of $N$ different response trajectories, $\{\tau_1, \tau_2, \ldots, \tau_N\}$. The total reward is calculated for each trajectory in the group. Following the standard GRPO procedure, these rewards are then used to compute a normalized advantage, $\bar{R}(\tau_i)$ is calculated for each trajectory in the group. Following the standard GRPO procedure, these rewards are then used to compute a normalized advantage.