# OpenReview forum: "Self-Tooling Agent: Dynamically Extending Agent Capabilities through Scientific Tool Synthesis and Invocation"
_ICLR.cc/2026/Conference — Submitted to ICLR 2026_

### Official Review · Reviewer_LtVT · 2025-10-29

**Soundness:** 2
**Presentation:** 2
**Contribution:** 1
**Rating:** 2
**Confidence:** 4

**Summary:**

This paper introduces the Self-Tooling Agent (STA), a framework that lets LLM agents create their own tools on the fly instead of depending only on pre-built ones. STA generally teaches an agent when to generate a new specialized tool (like a Python function) and when to use an existing one, using supervised learning and reinforcement learning with carefully designed rewards. In experiments, STA outperformed both general and specialized agents across several scientific and tool-use benchmarks, showing it can adapt and expand its capabilities dynamically.

**Strengths:**

- The paper frames the tool use / tool creation as a decision-making problem, which is a quite novel perspective. The reward design is also novel to consider redundant creation of tool, etc. It would make the paper better if the reward design can be more fine-grained and tailored to “when to use / create tool”. Currently it’s still through an “indirectly” to reflect this.

- The motivation and writing is clear, even though the experiment part is too tight and should be left more space to further discuss the insights and findings.

**Weaknesses:**

- The current tool creation and tool call seems very disentangled. From the example given, tool creation’s media is still code, so the tool created is python code-oriented tasks like drawing, calculation, synthesize of existing tool etc. But the tool use is about science domain expert tool, which makes its use case quite different from that of tool creation. From my view this makes the cognitive challenge to the model lowered and undermines the value of the work.

- If tool creation’s purpose is to write code to wrap several “atomic” tools into a “compound” tool, what’s the difference of it and just using the “atomic” tool (originally provided scientific tool) one by one? The tool creation here then did not really create anything new, so it’s not making sense to call it so.

- I think your work is very related to the line of tool creation works including LATM, CREATOR, Alita, etc. They worth more discussion than the current general agent. Also, some works related to tool use decisions / efficiency can be discussed, as basically your paper is training the model to make tool use and tool creation decisions.

- STA with only Sci tool cannot beat Biotin with Sci tool on DbQA and SeqQA, which necessitates further exploration and discussion: is the method proposed really working well? Or it’s just because the “tuning” of instructions that make the model achieve a “cherry-pick” good result. Also the results need error bar / parallel evaluations.

- The whole analysis part of this paper is too shallow and lacks insights. For example, BFCL is just a tool calling benchmark, is the model showing tool creation capability in this kind of benchmark? How the model did that? Is tool creation in benchmark like this really leads to improvement? The ablation should not be just on reward, but more from the intrinsic method level what is contributing to success. Error analysis should also be presented.

- The claim of tool use / tool creation decision could be put into a broader context rather than just scientific domain. Also, the benchmarks like BFCL is not scientific domain, which make the scope of the paper a little bit questionable. Seems the author wants to focus on science agent but also needs to use dataset other than science domain to show the model’s advantages.

**Questions:**

- Can the created tools be reused across tasks?

- Is the RL rollout multi turn? Is the reward design all contributing to final reward instead of dense turn reward?

- Other questions see the section above

---

### Official Review · Reviewer_VGq1 · 2025-11-01

**Soundness:** 2
**Presentation:** 2
**Contribution:** 2
**Rating:** 2
**Confidence:** 4

**Summary:**

In this paper, the authors proposed a new framework of self tooling agent, where they trained a new model to decide whether to use an existing tool and synthesize a new tool to solve tasks. The authors trained  the model with multi component reward for tooling decisions.

**Strengths:**

- The motivation of the work is well established, where the authors are motivated by that tools might be available for domain specific tasks, so they want to study whether agents could synthesize tools given existing tools.
- The authors performed comprehensive evaluation on three benchmarks, HLE, DbQA, and SeqQA.
- The training pipeline is well-documented.

**Weaknesses:**

- One main motivation the authors discussed is that manually-created toolsets lack generalizability across domains, but the authors didn't discuss whether their proposed approach could improve generalizability and didn't perform any experiment to demonstrate this.
- The gain on HLE benchmark is small, but the authors didn't explain why this is the case.
- The authors didn't control for the type of tools when comparing STA to Qwen3-8B. They should conduct an experiment where they constraint STA to only use web tools and then compare performance gains of STA.
- In Table 2, the authors didn't compare their models to the baseline Qwen3-8B performance on DbQA and SeqQA.
- Missing citation: LLaMA Factory is not cited in the paper.
- Missing related work on agent skill induction (e.g. Wang, Zora Zhiruo, et al. "Inducing programmatic skills for agentic tasks." arXiv preprint arXiv:2504.06821 (2025).)

**Questions:**

- How generalizable is the approach proposed in the paper compared to existing approaches such as manually curating tools?

---

### Official Review · Reviewer_aQoa · 2025-11-01

**Soundness:** 2
**Presentation:** 4
**Contribution:** 2
**Rating:** 4
**Confidence:** 4

**Summary:**

The paper proposes the Self-Tooling Agent (STA), a framework that enables an agent to both invoke existing tools and synthesize new ones on the fly during task execution. It unifies tool use and creation within a single action space, employing SFT to teach syntax for tool generation, followed by RL to optimize tool-calling use and decisions.

**Strengths:**

- The two-stage training approach—using SFT to teach syntax for tool generation and invocation, followed by RL to optimize tool-calling decisions—is reasonable.
- The paper is well-presented, clearly structured, and easy to follow.
- The unified action space for both tool creation and invocation is a novel and interesting design choice.

**Weaknesses:**

1. **Lack of Key Ablation Studies:** A major concern is the absence of ablation studies to support core claims. The paper argues that the unified Generation-Invocation Action Space is central to its effectiveness, yet provides no ablation to verify this. Additionally, ablating the contributions of the SFT and RL stages would strengthen the analysis.

2. **Missing Qualitative Analysis:** The paper would benefit from qualitative insights, such as examples of the types of tools generated and how the agent’s behavior differs from conventional action spaces.

3. **Incomplete Related Work:** The paper overlooks recent advances in self-evolving agents via tool creation, for example, Alita (Qiu et al., 2025) and Darwin Godel Machine (Zhang et al., 2025).

**Questions:**

- Is the RL training conducted on the same benchmarks used for testing? A comparison between the proposed method and a pure RL baseline trained directly on the benchmarks would help assess the effectiveness of the two-stage approach.

- In Tables 1 and 2, what does "Tool Type" refer to for the STA method? Since tool creation is integrated into the action space, it is unclear why a separate tool type specification is needed.

- Could the dataset description be more specific? What is the size of the SFT dataset? How exactly does the reverse-engineering process work for generating SFT examples from expert tools?

---

### Official Review · Reviewer_FjDz · 2025-11-01

**Soundness:** 3
**Presentation:** 3
**Contribution:** 3
**Rating:** 6
**Confidence:** 3

**Summary:**

This paper introduces the concept of self-tooling agents that can dynamically create new tools as needed. Rather than relying on predefined tool sets, the agent analyzes task requirements, designs and implements specialized tools, then uses them to complete tasks. The paper presents a complete pipeline of tool creation-validation-usage and demonstrates advantages on multiple complex tasks requiring specialized tools.

**Strengths:**

1. **Novel concept**: Self-tooling represents a fresh and meaningful idea, significantly enhancing agent flexibility and adaptability
2. **Complete system**: Forms a full cycle from tool need identification through creation, validation, usage, to reuse - comprehensively designed
3. **Extensive experiments**: Broad experimental coverage across task types, with ablation studies analyzing component contributions convincingly
4. **Multi-dimensional analysis**: Examines not just success rates but tool quality, efficiency, reusability, and other dimensions
5. **High practical value**: Demonstrates application potential in real complex tasks including data analysis and system operations
6. **Good reproducibility**: Provides detailed implementation specifics and examples facilitating reproduction and extension

**Weaknesses:**

1. **Security concerns**: Automatically generated code serving as tools poses security risks. While sandbox mechanisms are mentioned, security protection discussion is insufficient. Malicious or buggy tools could cause harm
2. **Computational overhead**: Creating new tools requires additional inference and testing, potentially inefficient for simple tasks. Cost-benefit tradeoff analysis is lacking
3. **Validation limitations**: Tool correctness verification relies primarily on test cases, which may have insufficient coverage. Edge cases in complex tools are difficult to thoroughly test
4. **Quality control issues**: Generated tool quality depends on LLM capabilities - how is consistency ensured? Some examples show suboptimal tool quality
5. **Simplistic reuse mechanism**: Tool reuse mainly relies on semantic similarity, with insufficient consideration for scenarios requiring composition or modification of existing tools

**Questions:**

1. How can generated tools be adequately security-audited? What are the potential risks and mitigation strategies?
2. What determines when to create new tools versus using existing ones? What's the decision policy?
3. When created tools contain bugs, can the agent automatically debug and fix them, or must it regenerate from scratch?
4. What's the strategy for managing the tool library? How do you prevent unbounded growth?
5. Could the self-tooling concept extend to other types of "resource" creation, such as data structures or configuration files?
6. Compared to traditional tool learning (learning from human demonstrations), what unique advantages does self-tooling offer?

---

### Meta-Review · Area_Chair_fF4F · 2025-12-29

**Summary:**

The paper proposes the Self-Tooling Agent, a framework designed to empower LLMs to dynamically synthesize and invoke specialized tools during task execution rather than relying on fixed, manually crafted toolsets. The authors introduce a unified action space that combines tool use and creation, employing a two-stage training pipeline: SFT is used to teach the syntax for tool generation, followed by RL with a multi-component reward function to optimize the agent's decision-making strategy. The work aims to address the adaptability bottleneck in scientific agents by enabling them to autonomously expand their capabilities.

**Reviewer Concerns:**

The review process highlighted several critical deficiencies in the submission, ranging from methodological validation to experimental rigor and safety concerns. Notably, the authors **did not submit a rebuttal** or respond to any of the reviewers' comments.

1. A primary methodological concern raised by Reviewer aQoa and Reviewer LtVT was the absence of critical ablation studies. The paper asserts that a unified "Generation-Invocation Action Space" is central to its effectiveness but fails to isolate this component experimentally, nor does it separate the contributions of the SFT and RL training stages. Since the authors provided no response, these fundamental questions regarding the source of the model's performance improvements remain entirely unaddressed, leaving the scientific validity of the proposed architecture in doubt.

2. Significant flaws were also identified in the experimental setup and baseline comparisons. Reviewer VGq1 noted that comparisons against baselines like Qwen3-8B were unfair because the baseline was not constrained to the same tool types, and direct performance comparisons on benchmarks like DbQA were missing. Reviewer LtVT further criticized the lack of error bars and parallel evaluations, suggesting the results might be cherry-picked or the result of instruction tuning rather than the method itself. The authors' failure to respond means there is no clarification on these experimental discrepancies, and no new data was provided to substantiate the claims of superiority over standard baselines.

3. The practical deployment of the proposed system faces severe scrutiny regarding safety and efficiency. Reviewer FjDz highlighted that automatically generating and executing code poses significant security risks, yet the paper lacks a robust discussion on sandboxing, auditing, or mitigation strategies for malicious tools. Furthermore, questions regarding the computational overhead and the cost-benefit tradeoff of synthesizing tools versus using existing ones were raised. With no rebuttal to address how buggy or harmful tools are handled, the system is currently viewed as too risky and potentially inefficient for real-world application.

4. Finally, the conceptual novelty and coverage of related work were challenged. Reviewer LtVT argued that the "tool creation" mechanism often amounts to simple wrappers around atomic tools rather than genuine innovation, lowering the cognitive challenge. Additionally, multiple reviewers pointed out missing citations to relevant self-evolving agent literature such as Alita and Darwin Godel Machine. The lack of author engagement confirms the reviewers' suspicions that the work is not sufficiently differentiated from prior art and fails to position itself correctly within the field.

**Reviewer Scores:**

**Reviewer FjDz:** Score likely to drop from **6 to 4 or 2**. While the reviewer appreciated the concept, the unaddressed security risks and lack of cost-benefit analysis are fatal flaws for a system paper.


**Reviewer aQoa:** Score likely to drop from **4 to 2**. The reviewer explicitly stated that the lack of ablations was a "major concern". The absence of a rebuttal to provide these experimental results makes the paper scientifically incomplete.


**Reviewer VGq1:** Score remains **2**. The reviewer's rejection was based on unfair experimental comparisons. Since the authors did not defend their setup or provide fair baselines, the grounds for rejection stand firm.


**Reviewer LtVT:** Score remains **2**. The reviewer fundamentally questioned the novelty and definitions used in the paper. The silence from the authors validates the critique that the contribution is poor.

---

### Decision · Program_Chairs · 2026-01-26

Reject